# Quadrupolar excitons in MoSe$_2$ bilayers

Jakub Jasiński [1,2], Joakim Hagel [3], Samuel Brem [4], Edith Wietek [5], Takashi Taniguchi [6], Kenji Watanabe [7], Alexey Chernikov [5], Nicolas Bruyant [2], Mateusz Dyksik [1], Alessandro Surrente [1], Michał Baranowski [1], Duncan K. Maude[2], Ermin Malic [4] & Paulina Plochocka [1,2] ✉

The quest for platforms to generate and control exotic excitonic states has greatly benefited from the advent of transition metal dichalcogenide (TMD) monolayers and their heterostructures. Among the unconventional excitonic states, quadrupolar excitons—a superposition of two dipolar excitons with anti-aligned dipole moments—are of great interest for applications in quantum simulations and for the investigation of many-body physics. Here, we unambiguously demonstrate the emergence of quadrupolar excitons in natural MoSe$_2$ homobilayers, whose energy shifts quadratically in electric field. In contrast to trilayer systems, MoSe$_2$ homobilayers have many advantages, which include a larger coupling between dipolar excitons. Our experimental observations are complemented by many-particle theory calculations offering microscopic insights in the formation of quadrupolar excitons. Our results suggest TMD homobilayers as ideal platform for the engineering of excitonic states and their interaction with light and thus candidate for carrying out on-chip quantum simulations.

Two-dimensional (2D) layered semiconductors, such as transition metal dichalcogenides (TMDs), have emerged as an ideal playground to study exciton physics on the nanoscale, essentially due to the intricate valley physics and the greatly enhanced electron–hole attraction related to the reduced dimensionality and dielectric screening in the monolayer limit[1–7]. The subsequent development of van der Waals heterostructures significantly enriched this field of research. The absence of the lattice-matching constraints for TMDs and many other emerging layered 2D materials opens a new paradigm in material engineering, where different materials can be seamlessly stacked into virtually limitless combinations, whose properties can be tuned by both the material selection and the relative orientation[8–11].

For instance, homobilayers and heterostructures support long-lived dipolar interlayer excitons (IXs), where electrons and holes reside in different layers[12–18], and hence can be can be easily tuned by external electric field[19–22]. Transition metal dichalcogenide heterostructures have emerged as an excellent solid-state platform for exploring many-body physics and quantum phases arising from monopolar[23] and dipolar interactions[24–33], entering fields traditionally dominated by ultracold atoms[34–38]. Very recently it has been demonstrated that TMD heterostructures can also host more complex quasiparticles, referred to as quadrupolar excitons[39–46], stemming from the hybridization between two dipolar excitons with opposite dipole moments. The higher-order symmetry causes the quadrupole-quadrupole interactions to be substantially different compared to the dipole–dipole ones. In particular, their non-local interactions can be finely tuned by the application of electric field. The quadrupolar interactions enable new collective phenomena beyond monopolar and dipolar interactions such as the exotic rotons, new flavors of Bose-Einstein condensate, charge density wave or topological superfluids[44,47–50]. The solid-state

[1]Department of Experimental Physics, Faculty of Fundamental Problems of Technology, Wroclaw University of Science and Technology, Wroclaw, Poland. [2]Laboratoire National des Champs Magnétiques Intenses, EMFL, CNRS UPR 3228, Université Grenoble Alpes, Université Toulouse, Grenoble and Toulouse, France. [3]Department of Physics, Chalmers University of Technology, Gothenburg, Sweden. [4]Department of Physics, Philipps-Universität Marburg, Marburg, Germany. [5]Institute of Applied Physics and Würzburg-Dresden Cluster of Excellence ct.qmat, Technische Universität Dresden, Dresden, Germany. [6]Research Center for Materials Nanoarchitectonics, National Institute for Materials Science, Tsukuba, Japan. [7]Research Center for Electronic and Optical Materials, National Institute for Materials Science, Tsukuba, Japan. ✉e-mail: paulina.plochocka@lncmi.cnrs.fr

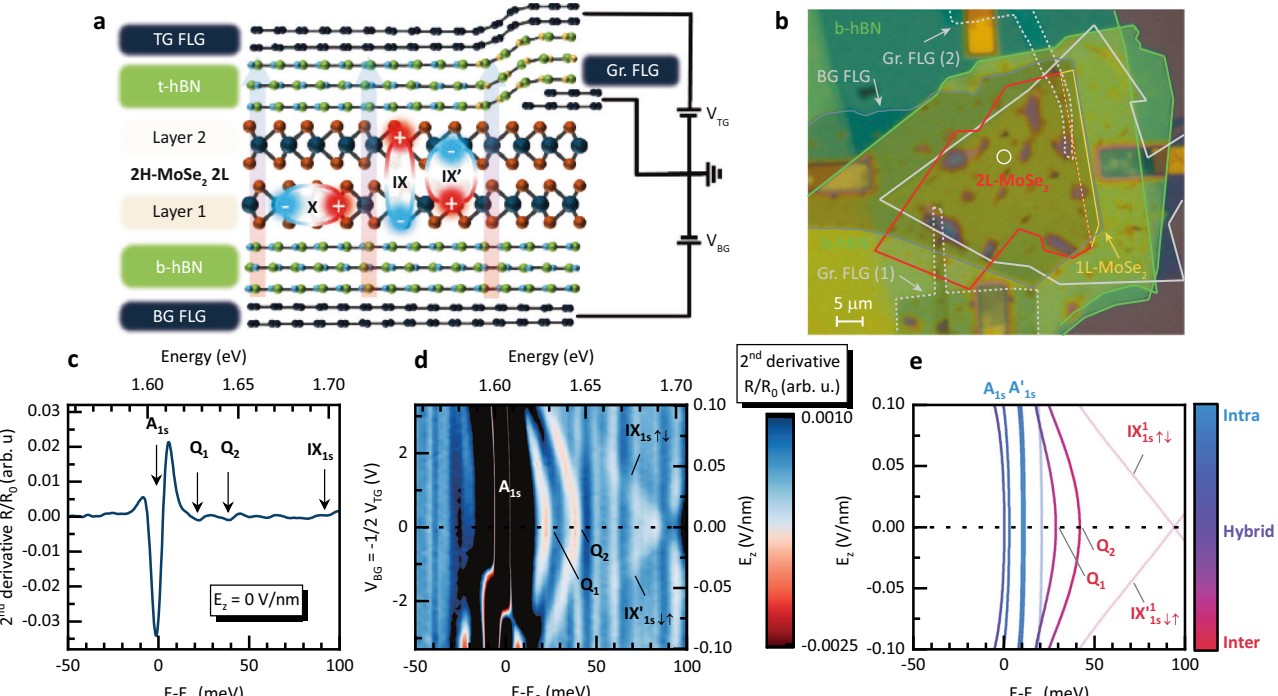

**Fig. 1 | Natural MoSe₂ bilayer device and optical response under applied out-of-plane electric field. a** Schematic of the device. 2H-stacked natural bilayer of MoSe₂ encapsulated by insulating bottom and top hBN layers. Few layer graphene (FLG) serve as the bottom (BG) and top (TG) gates, respectively, forming a capacitor-like structure. Two additional FLG flakes physically touch the MoSe₂ bilayer for grounding purposes (Gr. FLG (1)/(2)). The effect of an applied out of plane electric field ($E_z$ indicated by the arrows) is shown on the intralayer exciton (X), and the interlayer excitons with opposite dipole moments (IX and IX'). **b** Micrograph of the sample. The white circle indicates the measurement spot. **c** Second derivative of normalised reflectivity ($R/R_0$) at $E_z = 0$ V/nm. **d** False-color map of the 2$^{nd}$ derivative of reflectivity as a function of the gate voltages (left axis) and the corresponding $E_z$

(right axis). The intensity of the strongest neutral $A_{1s}$ exciton transition is intentionally saturated to reveal the behaviour of the weaker interlayer transitions. Note, that at negative electric fields, $E_z < -0.5$ V/nm, the device is unintentionally doped, thus for simplicity, we focus on the analysis of the positive $E_z > 0$ electric field data. **e** Calculated evolution of the excitonic energy landscape under the influence of the electric field. The color scale corresponds to the spatial character of the excitons i.e. intralayer (blue), hybrid (purple) or interlayer (red). The opaque (semi-transparent) lines correspond to the spin-singlet (spin-triplet) states. The bottom energy scales ($E - E_0$) in panels **c** and **d** are shown with respect to the $A_{1s}$ exciton energy $E_0 = 1.606$ eV.

matrix endows the quadrupolar states with robustness, which instead eludes their molecular counterparts[51,52], and is particularly attractive to implement quantum simulation protocols and to reveal unconventional quantum states, many body phases[48–50], and phase transitions[44,45]. The multipolar character of excitons in TMD structures can be continuously tuned between quadrupolar and dipolar states due to the nonlinear Stark effect[39,40] or mixing with other excitonic species (as we show herein), which enables a continuous control over many-body interactions.

So far the formation of quadrupolar states in TMD systems has been explored (both theoretically and experimentally) only for TMD heterotrilayers[39–46], which enforce the formation of interlayer excitons with anti-aligned static dipole moments. Similar conditions can be found in TMD homobilayers[15,17,53,54], suggesting that quadrupolar excitons might also form in these structures. However, they have remained elusive so far.

Here, we demonstrate the existence of quadrupolar states in a natural 2H-stacked homobilayer of MoSe₂. In our double gated device we identify two types of interlayer transitions with dipolar and quadrupolar character. Combining many-particle theoretical modelling, and electric field dependent reflectivity measurements, we provide a microscopic understanding of the complex excitonic landscape in a natural MoSe₂ bilayer. We show that the quadrupolar states emerge from the coupling between the dipolar transitions. The observed excitonic states, including the quadrupolar excitons, can be effectively tuned, with the use of electric field, between interlayer, hybrid and intralayer character showing that natural MoSe₂ bilayers are promising

candidates to study many-body physics driven by field-tunable electric multipolar interactions.

## Results and Discussion
### Observation of quadrupolar excitons
We have investigated a natural 2H-stacked MoSe₂ homobilayer, fully encapsulated in hexagonal boron nitride (hBN). The hBN encapsulated MoSe₂ is grounded by a few layer graphite (FLG) electrodes. Two additional FLG electrodes are used as the top and the bottom gates. Schematic and microscope images of the device are shown in Fig. 1a, b. During the fabrication of the device by the dry-transfer method, the stack was annealed after each stamping step. The goal of annealing was to minimize the concentration of bubbles and simultaneously improve the adhesion between the consecutive layers (see Methods section for more details on the fabrication procedure). To reveal the complex excitonic landscape of the natural MoSe₂ bilayer, we studied its optical response as a function of the out-of-plane electric field ($E_z$) using the capacitor-like design of the structure which allows for the independent control of the out-of-plane electric field and carrier doping (see Methods for details).

A typical reflectivity spectrum of bilayer MoSe₂ (measured at temperature of 5 K), shown as a second derivative is presented in Fig. 1c. The spectrum is dominated by a strong resonance related to the $A_{1s}$ exciton state, accompanied on the high energy side by three weaker transitions, labelled as $Q_1$, $Q_2$ and $IX_{1s}$. To understand the origin of these transitions, we track their evolution as a function of the electric field. In Fig. 1d we present the reflectivity spectrum at varying

electric field, plotted in the form of a false-color map. Characteristic features can be identified in the false colour map, providing deeper insight into the exciton landscape and the mutual interaction of the excitonic states. The states labelled as $IX_{1s,\uparrow\downarrow}$ and $IX'_{1s,\downarrow\uparrow}$ exhibit a linear Stark shift, consistent with their dipolar, interlayer character[54–57]. Matching the observed shift to the Stark shift simulated with the model detailed below, we estimate the dipole moment to be $d \simeq 0.5–0.6$ e nm, which is in the range of the dipole length reported for other $MoSe_2$ bilayers[55,56]. In addition, the new states labelled $Q_1$ and $Q_2$ exhibit a distinct, quadratic Stark shift at low electric fields. This behaviour is the unequivocal evidence of their quadrupolar nature[39,40,43], which stems from the coupling of a pair of anti-aligned dipolar states. The symmetric arrangement of charges in an electric quadrupole yields a zero dipole moment at $E_z = 0$. However, increasing electric field displaces the charges, and the quadrupolar state gradually acquires a dipole moment, giving rise to the non-linear Stark shift. To corroborate the assignment of the $Q_1$ and $Q_2$ as quadrupolar excitons, we also plotted in Fig. S1 the electric field dependence of the static electric dipole moment, calculated as $\frac{dE}{dE_z}$. The non-linear dependence of the quadrupolar exciton energy on the electric field translates to a vanishingly small electric dipole at low fields. The static dipole moment increases with increasing electric field and steadily approaches the dipole moment obtained for the dipolar spin-triplet interlayer exciton $IX^A_{1s\uparrow\downarrow}$. At higher fields, around 0.1 V/nm, one can observe the deviation from the expected behaviour which stems from the interaction with other excitonic states as we discuss in the further part of the manuscript. Additionally, analogous excitonic resonances shifting quadratically with the applied electric field, which attests to their quadrupolar nature, were observed on different spots on the main device (Fig. S2) and also on a 2nd device shown in Fig. S3.

## Microscopic model

To provide a detailed microscopic understanding of our observations, we complement our experiments with an effective many-particle model that allows for the identification of the key coupling mechanisms. The symmetric band structure in naturally stacked bilayers hosts a fourfold degeneracy, stemming from the combination of valley and layer degeneracy. This applies to all intralayer and interlayer exciton species, including the A-exciton, interlayer spin-singlet ($\uparrow\uparrow$, $\downarrow\downarrow$) and spin-triplet states ($\uparrow\downarrow$, $\downarrow\uparrow$) (see Supplementary Fig. S7(a,b,c)). For instance, the spin-singlet interlayer excitons (Supplementary Fig. S7(b)), $IX_{\uparrow\uparrow}$ and $IX_{\downarrow\downarrow}$ have a reversed dipole moment, i.e., exchanged positions of the electron and hole, with respect to the other two ($IX'_{\uparrow\uparrow}$ and $IX'_{\downarrow\downarrow}$), leading to their mixing via the dipole exchange interaction (see Section II in SI for further details). The effective Hamiltonian, which includes all possible interaction channels, can then be written as

$$H = H_0 + H_T + H_{QC}.$$

Here $H_0$ describes the electron and hole Coulomb interaction through the generalized Wannier equation,[58] together with the exciton response to the external electric field. $H_T$ is the tunneling contribution, which takes into account both electron and hole tunneling[59]. The last term $H_{QC}$ contains the effective dipole exchange coupling $\tilde{J}$ (see Eq. (S5) in SI), giving rise to the formation of quadrupolar excitons.

We initially focus on the exchange coupling to explain the nonlinear shift of the $Q_1$ and $Q_2$ transitions. We assume that $\tilde{J}$ only mixes the 1s interlayer exciton states of the spin-singlet configuration $IX_s$ ($\uparrow\uparrow$, $\downarrow\downarrow$) and opposite dipole moments. This tentative assumption is motivated by the more pronounced signature of Q-states compared to IX features, which suggests a higher oscillator strength characteristic for singlet transition. Moreover, the mixing is assumed to stem from the Coulomb interaction, which is a spin-conserving interaction. Such mixing between the necessary spin-triplet states would not be spin-

conserving in a bilayer system. Nevertheless, qualitatively similar quadrupole formation could be expected assuming an equally efficient coupling between interlayer triplet states. We infer that four spin-singlet IXs mix through two possible interaction paths ($\tilde{J} = J + J'$) (See Eq. (8) in SI) schematically shown in Fig. 2a–c. The first path couples anti-aligned dipolar IXs corresponding to the same valley but with opposite spin configuration ($IX_{\uparrow\uparrow} + IX'_{\downarrow\downarrow}$ and $IX_{\downarrow\downarrow} + IX'_{\uparrow\uparrow}$), schematically drawn in Fig. 2a, c and denoted as $J$. The second path, indicated as $J'$, mixes anti-aligned dipolar IXs localized in the opposite valleys, but with the same spin configuration ($IX_{\uparrow\uparrow} + IX'_{\uparrow\uparrow}$ and $IX_{\downarrow\downarrow} + IX'_{\downarrow\downarrow}$), as schematically represented in Fig. 2b, c.

The evolution of the energy landscape of IXs under electric field in the absence and in the presence of the exchange couplings $J//J'$ is schematically presented in Fig. 2d–f. When the $J//J'$ couplings are not accounted for (Fig. 2d), the application of an electric field gives rise to two linearly shifting IX states: the higher energy spin-singlet and the lower energy spin-triplet states, offset by the spin-orbit coupling in the conduction band. The inclusion of the first term $J$ in Fig. 2e mixes the spin-singlet IXs, which yields two quadrupolar excitons with opposite curvature, i.e., the (symmetric – red-shifting) Q and (anti symmetric – blue-shifting) Q' separated by an energy $2J$ at zero electric field (and by $\pm J$ from the IX singlet states in the non-interacting picture). The $J'$ coupling, added in Fig. 2f, leads to a further splitting of the quadrupolar branches into $Q_1$ and $Q_2$, and $Q'_1$ and $Q'_2$, each pair separated by $2J'$ (see the detailed model description in Section II in the SI). As the lower energy spin-triplet IXs ($IX_{\uparrow\downarrow}$, $IX_{\downarrow\uparrow}$, $IX'_{\uparrow\downarrow}$, $IX'_{\downarrow\uparrow}$) are not spin-conserving, they remain unaffected by the $J//J'$ coupling and thus they follow the standard linear Stark shift.

By matching the values of $J$ (energy de-tuning from the spin-triplet IX plus spin orbit coupling of ~ 22 meV[60] at $E_z = 0$ V/nm) and $J'$ (half of the separation between the $Q_1$ and $Q_2$ at $E_z = 0$ V/nm) to match the experimentally observed redshifting quadrupole branch, we obtain $J = 90$ meV and $J' = 8$ meV. The results of the simulation are shown in Fig. 1e. The good qualitative agreement with the experiment summarized in Fig. 1d demonstrates that our effective Hamiltonian successfully explains the experimentally observed electric field-induced nonlinear energy shift of the $Q_1$ and $Q_2$ states. These states, formed from linear combination of spin-singlet IXs ($Q_1 \sim IX'_{\uparrow\uparrow} + IX_{\downarrow\downarrow}$ and $Q_2 \sim IX'_{\downarrow\downarrow} + IX_{\uparrow\uparrow}$), correspond to symmetric quadrupole branches which red shift with increasing electric field. According to the presented analysis the IX states ($IX_{\uparrow\downarrow}/IX'_{\downarrow\downarrow}$) with resonance around 90 meV above the $A_{1s}$ at $E_z = 0$ V/nm are transitions originating from optically bright spin-triplet states[61]. Due to the lack of coupling $\tilde{J}$ these preserve their dipolar character exhibiting a linear Stark shift. Here we note that the opposite assignment of the dipolar transition origin can also be found[54]. Unfortunately, our model does not allow for a definitive differentiation between a singlet or triplet origin of quadrupolar states. Importantly the singlet or triplet nature of the interlayer is not essential to interpret the quadratic shift of $Q_1$ and $Q_2$ transitions.

## Charge tunneling

To reveal the importance of the charge tunneling mediated interaction between different excitonic species we plot the measured reflectivity spectra over an extended energy and electric field ranges as shown in Fig. 3a, b. At higher electric field we observe an anti-crossing behaviour of $A_{1s}$ with interlayer exciton species such as the spin-triplet $IX_{1s,\uparrow\downarrow}$, 2s spin-singlet $IX_{2s,\uparrow\uparrow}$ and the quadrupolar $Q_{1/2}$ states as indicated by green dashed lines in Fig. 3b. To explain this behaviour we incorporate the electron tunneling term ($t_e$) into our model (schematically shown in Fig. 2a, b by the arrows labeled $t_e$). In Fig. 3c we present the results of the simulation, taking the electron tunneling into account, which clearly shows that the anti-crossing behaviour is well captured by the model (see also Fig. S8 in SI showing progressively the contribution of the various coupling mechanisms to the exciton spectrum). Note that electron tunneling is usually considered symmetry-forbidden in

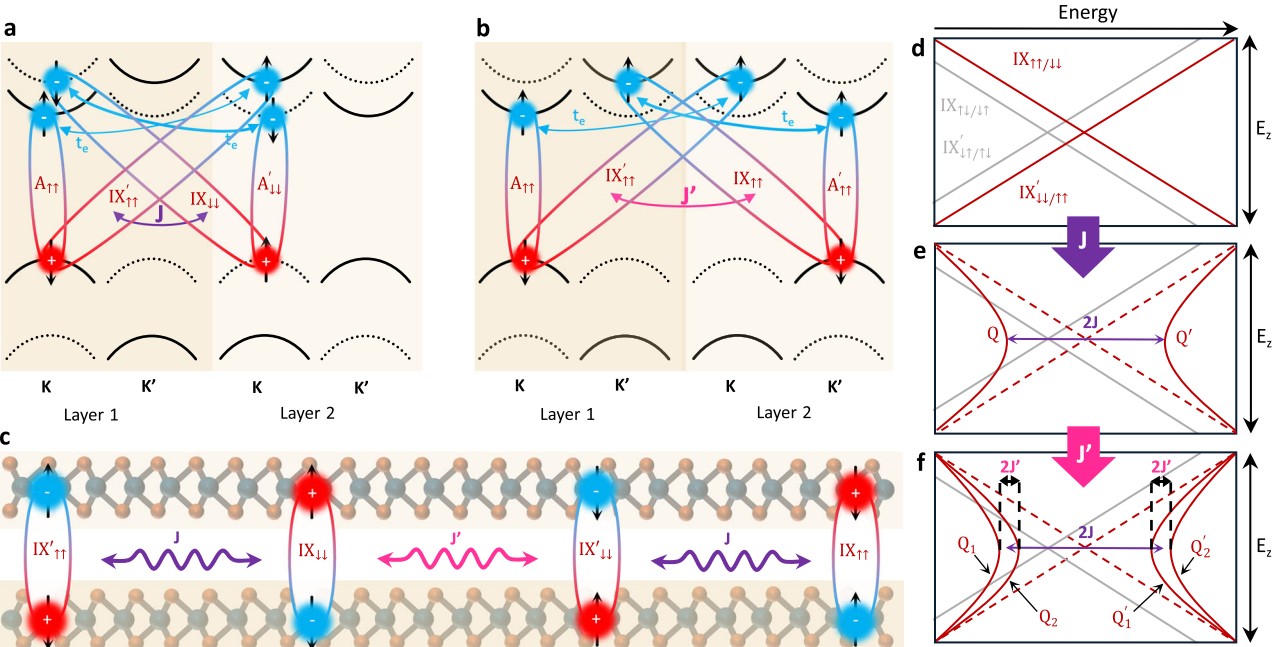

**Fig. 2 | Dipolar exchange interaction in natural MoSe₂ bilayer. a** Schematic of the $J$ coupling between one of the oppositely aligned pairs of singlet IXs (same valley, different spin) and the interplay with electron tunneling to their respective A excitons. **b** Schematic of the $J'$ coupling between one of the oppositely aligned pairs of singlet IXs (different valley, same spin) and the interplay with electron tunneling to their respective A excitons. **c** Schematic of the real space interaction of the four degenerate singlet IXs via $J$ and $J'$ couplings. **d** Scheme of IX exciton species energy as a function of the electric field for the singlet (↑↑, ↓↓) and triplet (↑↓, ↓↑) IX species without $\bar{J}$ coupling. **e** including the $J$ coupling term IX singlet states mix forming quadrupole branches Q and Q′. **f** exciton landscape including both $J$ and $J'$ coupling which splits degeneracy of Q and Q′ branches. The dashed lines in **e** and **f** correspond to spin-singlet states in the non-interacting picture **d**, from which the quadrupolar branches formed.

naturally stacked homobilayers ($H_h^h$ stacking)[62]. Nevertheless, our results demonstrate that some electron tunneling occurs, and it is crucial for the correct description of the excitonic landscape in MoSe₂ bilayers under the electric field. In the simulation, we assume the electron tunneling term to be $t_e = 11.9$ meV, corresponding to the calculated value for the $H_h^h$ stacking[59]. The much stronger hole tunneling term ($t_h = 56.3$ meV[59]) predominantly drives the hybridization between various interlayer and intralayer states[53,63]. These manifest in the exchange of the oscillator strength with increasing electric field as they approach energetically (see also the extended energy range data in Supplementary Fig. S4b, c, where the observed hole tunneling mediated hybridizations are marked).

Another significant effect, stemming from the hybridization due to charge tunneling (both electron and hole), is reflected in the change of the intra-inter layer character of excitonic states, which is tuned by the value of electric field. This is shown in the simulated spectra of Fig. 3c as the color coding of the lines, where the blue, purple and red correspond to the intralayer, hybrid and interlayer character, respectively. For example, the electron tunneling changes the character of quadrupolar excitons when they approach the $A_{1s}$ transition with increasing electric field. Around the anti-crossing region, the quadrupoles rapidly change their character from interlayer to intralayer, with a negligible energy shift as a function of the electric field. At the same time, some of the A excitons acquire a partially interlayer character. This is in contrast to the heterotrilayer case, where the quadrupolar exciton is the lowest state of the system, and its energy shift steadily approaches a rate which is characteristic for the dipolar interlayer exciton[39,43]. Similar electric field induced change of exciton spatial characters can be observed for other transitions (see also Fig. S4c in the Supplementary Information). The charge carrier tunneling also explains the suppression of the blue-shifting anti-symmetric quadrupole branch in our spectra as a result of mixing with $A_{2s}$ exciton states. The antisymmetric exciton branch is expected to be at energies very

close to the $A_{2s}$ intralayer exciton at zero field. Due to this close proximity, the antisymmetric quadrupolar excitons hybridize with intralayer $A_{2s}$ exciton, and their signature in the optical spectrum vanishes. However, the mixing of these intra- and interlayer states manifests as a splitting of $A_{2s}$ exciton at zero electric field. This state is not expected to exhibit any splitting in the non-interacting picture, but in the presence of quadrupolar exciton complexes, it shows a finite splitting, following hybridization with interlayer excitons such as $IX_{2s,↑↑}$ or the antisymmetric branch of the main quadrupolar excitons. Consequently, the $A_{2s}$ state splits into $A_{2s}$ and $A_{2s}'$, even at zero electric field, as can be seen both in the experimental (Fig. 3b) and theoretical (Fig. 3c) spectra (see also Supplementary Fig. S8, where the influence of the individual couplings is shown). The conclusions drawn from the reflectivity spectra are supported by PL measurements shown in Fig. S5. In the evolution of the PL spectra in electric field, the characteristic anti-crossing behaviour when quadrupolar states approach the $A_{1s}$ exciton can be observed, together with the red shift of the $A_{2s}$ emission.

## Discussion

The origin of the couplings $J/J'$ has so far been attributed to low density effects in the Coulomb Hamiltonian. As demonstrated, the exchange coupling does indeed lead to terms which mix different dipoles and qualitatively fits well within the picture that $J \gg J'$. This is since $J$ includes both long and short range electron-hole exchange, whereas the short-range interaction is symmetry forbidden in $J'$ due to the mixing of different valleys (see expression for $H_{QC}$ in Eq. (S8) in SI)[64]. For intralayer excitons, this coupling has been calculated to be around 20 meV[65] and is expected to be smaller for interlayer excitons due to the reduced wave function overlap between the layers. Other effects such as density dependent dipole–dipole attraction might play a role in enhancing the mixing between different dipoles. Taking into account the device-to-device variability of the properties of TMD-

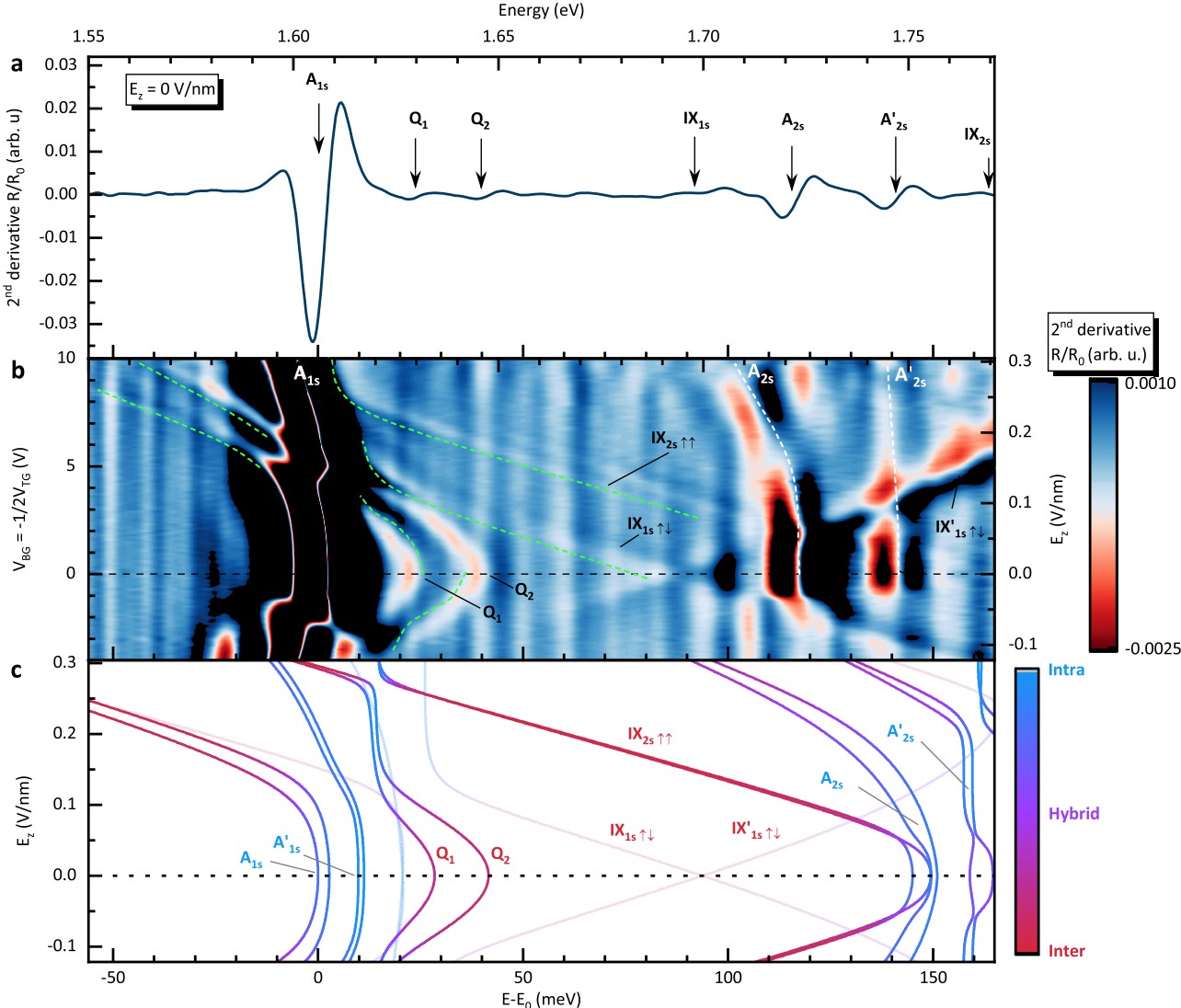

**Fig. 3 | Extended exciton energy landscape under electric field. a** Second derivative of reflectivity without electric field. **b** False-color map of $2^{nd}$ derivative of reflectivity as a function of applied gate voltages (left axis) and the corresponding $E_z$ (right axis). **c** Simulated exciton energy landscape in electric field which includes the hole and electron tunneling as well as the coupling $\bar{J}$ that forms the quadrupoles. The color scale corresponds to the spatial character of the excitons i.e. intralayer (blue), hybrid (purple) or interlayer (red). The opaque (semi-transparent) lines correspond to the spin-singlet (spin-triplet) states. The bottom energy scales ($E - E_0$) in panels **a** and **b** are shown with respect to the $A_{1s}$ exciton energy $E_0 = 1.606$ eV.

based devices, the clear experimental signatures of exciton quadrupole formation, together with the overall good qualitative agreement with theory, indicate that the coupling between the different dipoles are much stronger than previously thought. Our effective Hamiltonian successfully explains the evolution of the exciton landscape in bilayer $MoSe_2$. The formation of a quadrupolar state is driven by the dipolar exchange interaction between interlayer spin-singlet states. At the same time, the interlayer triplet states preserve their dipolar character, shifting linearly in the electric field. In addition, the hole and electron tunneling are responsible for the observed avoided crossing behaviour, and hybridization of the states.

In summary, we have investigated the evolution of the exciton energy landscape under external electric field in natural $MoSe_2$ homobilayers. Notably, for the first time, we observe quadrupolar exciton states in a natural $MoSe_2$ bilayer. These excitonic transitions, characterized by nonlinear shift in an electric field, exhibit a much stronger dipolar exchange interaction than the one observed in heterotrilayers. Our experimental observations are accurately captured by the proposed many-particle effective Hamiltonian. We propose that

dipolar excitons are characterized by spin-triplet configuration, while quadrupolar states emerge from the exchange coupling of the interlayer spin-singlet excitons. Moreover, our model highlights the importance of hole and electron tunneling for understanding the exciton landscape evolution under the electric field.

Our research underscores the potential of $MoSe_2$ bilayers to serve as a field-tunable exciton playground, wherein the mutual interaction of exciton states facilitates the effective tuning of their spatial and electric multipole characteristics via electric fields. Therefore, we show that natural $MoSe_2$ bilayers display potential to be considered as a solid-state platform to study many-body physics driven by field-tunable electric multipolar interactions. The inherent robustness of a homobilayer as compared to layer-by-layer stacking of TMD heterobi- or trilayers makes the platform proposed here easier to incorporate reliably into devices. This stems from the fact that homobilayers are not prone to imperfect flake alignment, flake rearrangement during deposition and possible post-stacking surface reconstruction for lattice commensurate stacks, which unavoidably plague other heterostructures.

## Methods

### Sample fabrication

The sample was fabricated using mechanically exfoliated flakes and stacked one by one using the dry transfer method. Each step of layer deposition was followed by annealing in ambient conditions, by ramping the temperature from 100° to 150° for the duration of ~15 min. At the final step, the sample was annealed for ~15 min at 200°. The goal of the annealing was to remove or coagulate air bubbles that notoriously form in TMD stacks during dry transfer deposition. The $MoSe_2$ bilayer is encapsulated with hBN and sandwiched in between few layer graphite (FLG) layers acting as the bottom and top gates. Two additional FLG layers are connected directly to (physically touching) the $MoSe_2$ bilayer serving as the grounding contacts, one as a spare contact. All FLG contacts overlap the nearby evaporated gold paths through which the voltage is applied. The gold pads are pre-deposited on the $SiO_2$ substrate. Our sample design does not require any additional lithography after the stack is transferred on the substrate, which minimizes the risk of introducing defects and lower the optical quality of the sample.

### Measurements

The sample is wire bonded in a chip carrier installed in a custom-made electrical adapter for the cold finger inside a helium flow cryostat. All presented measurements were performed at cryogenic temperatures of ~5 K.

We characterized the influence of the applied gate voltages at the bottom ($V_{BG}$) and top ($V_{TG}$) gates. To adjust for the unequal thicknesses of the bottom and top insulating layers of hBN, we found the optimal gate voltage ratio which minimizes the effect of free carrier doping during electric field sweep to be $V_{BG} = -\frac{1}{2}V_{TG}$. To make sure that we keep the most neutral doping level we checked also the ratio of neutral to charged exciton (trion) by applying gate voltages of the same polarity ($V_{BG} = \frac{1}{2}V_{TG}$). We found the $V_{BG} = \frac{1}{2}V_{TG} = 0$ V to be the optimal initial voltages due to the highest ratio of neutral to charged exciton, both in PL and Reflectivity (Supplementary Fig. S6). The reflectivity measurements were performed using a Tungsten-Halogen white light source, while PL used a 532 nm continuous wave laser at ~1 mW power.

### Data analysis

The details of the analysis of the reflectivity spectra are described in section III of the Supplementary Information. Fig. S9 shows the scheme of the data processing. The comparison of the reflectivity spectra in the form of $R/R_0$ and its 1st and 2nd derivatives are shown in Fig. S10.

The strength of the applied electric field was calculated by matching the dipole moments of the measured and simulated spin-singlet interlayer exciton for low electric field/gate voltages, far from the crossing region.

### Theoretical model

Exciton energies were modelled using an effective many-particle theory based on the density matrix formalism and input from density functional theory[57]. A two-particle tunneling Hamiltonian is formulated and the excitonic response to the electric field is included to first order[53]. The exchange interaction giving rise to the quadrupole formation is included from the low-density Coulomb Hamiltonian and matching the coupling strength to the experiment.

## Data availability

The experimental and theoretical datasets generated and/or analysed during this study are available at https://doi.org/10.5281/zenodo.14584540.

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

## Acknowledgements

J.J. acknowledges funding from the National Science Centre Poland within the Preludium Bis 1 (2019/35/O/ST3/02162) program. M.B. acknowledges funding from the National Science Centre Poland within the Sonata Bis (2020/38/E/ST3/00194) program and OPUS LAP (2021/43/I/ST3/01357). All authors thank Marzia Cuccu and Sophia Terres for their support in the laboratory work. P.P acknowledge supported through the EUR grant NanoX no. ANR-17-EURE-0009 in the framework of the "Programme des Investissements d'Avenir". The Marburg group (S.B. and E.M) acknowledges funding from the Deutsche Forschungsgemeinschaft (DFG) via SFB 1083 (project B9). K.W. and T.T. acknowledge support from the JSPS KAKENHI (Grant Numbers 21H05233 and 23H02052) and World Premier International Research Center Initiative (WPI), MEXT, Japan. A.C. and E.W. gratefully acknowledge funding from the Deutsche Forschungsgemeinschaft via SPP2244 grant (Project-ID: 443405595) and the Würzburg-Dresden Cluster of Excellence on Complexity and Topology in Quantum Matter (ct.qmat) (EXC 2147, Project-ID 390858490). E.M. and A.C. acknowledge DFG funding via project 542873285.

## Author contributions

J.J. has carried out all optical experiments and drafted the text and figures of the main manuscript and the supplementary information. J.H. and S.B. under the supervision of E.M. developed the theoretical model and performed the simulations. E.W. and A.C. have provided the necessary training and participated in the fabrication of electrical devices and electrical measurements. T.T. and K.W. have provided the high-quality hexagonal boron nitride for encapsulation of the sample. N.B. has been involved in the optimization of the experimental setup for electrical measurements and involved in those measurements. M.D.A.S., D.K.M. contributed to data analysis, interpretation of results

and manuscript preparation. M.B. and P.P. have proposed the goal of the scientific inquiry, the methodology and refined the manuscript text and the interpretation of the experimental results.

## Competing interests

The authors declare no competing interests.
