## [Transparent Peer Review file · Nature Communications]

Quadrupolar Excitons in MoSe2 Bilayers

Corresponding Author: Dr Paulina Plochocka

Version 0:

Reviewer comments:

Reviewer #1

(Remarks to the Author)

The authors have included additional experimental details and analyses, which strengthen their claims for publication in Nature Communications. A minor suggestion is to indicate the measured positions of the sample in Figure 1b, as there are numerous bubbles present in their device.

Reviewer #2

(Remarks to the Author)

I thank the authors for their rebuttal and revised manuscript. I have the following comments:

1. I agree with Referee 1 about the quality of the data being far from ideal. Nevertheless, the quadrupolar exciton is conclusively observed, so although the quality is unfortunate, the experiment evidence is present for the QX.
2. I have mixed opinions about the “novelty” or “excitement” of the discovery of QX in a bilayer system. I actually think the discovery is very nice, but I believe it is oversold in the revised manuscript. I guess I differ from Referee 3 who asked for more comments about why this is useful and therefore stronger statements about quantum simulation etc are now more prominent. Perhaps the intro could introduce QX more generally with these applications in mind, rather than making the specific comments about this particular bilayer QX? Softening the claims about the “usefulness” would be more honest in my view. I note also that multiple groups have manufactured the trilayer structures without too much difficulty, so I don't think that the fabrication challenge is such a great selling point.
3. The authors have not answered my original query (#2) about the lack of magneto-optics experiments which allows more clear assignment of the spin and valley configurations of the excitons. At the moment, I believe the authors have made a significant error in their interpretation due to the lack of such an experiment: their entire model and simulation rests on the claim that the QX arise from spin singlet states and the IX is a spin triplet. This is directly in contradiction to Nature Comms 15 4377 (2024), which provides strong experimental evidence that the IX is a spin singlet. The authors entire model / simulation is based on false starting point, as far as I can ascertain.

Based on my comment #3, in spite of the novelty of the QX in a bilayer system, I do not find the manuscript suitable for publication.

Reviewer #3

(Remarks to the Author)

I reviewed the author's response and manuscript revisions. The manuscript has been improved considerably and a transparent description of the experimental method and data analysis has been included. I agree with Reviewer 2 that magneto-optical experiments or other additional evidence could reduce uncertainty in the assignment of spin configurations and the modelling, which would constitute an important improvement. Nevertheless, I believe the main point of the manuscript, the observation of quadrupolar excitons in MoSe2 bilayer, is compelling and I therefore recommend the manuscript for publication in Nature Communications.

RESPONSE TO REVIEWERS' COMMENTS

We acknowledge that the referees appreciate our efforts to improve our manuscript.

Reviewer #1 (Remarks to the Author):

The authors have included additional experimental details and analyses, which strengthen their claims for publication in Nature Communications. A minor suggestion is to indicate the measured positions of the sample in Figure 1b, as there are numerous bubbles present in their device.

We would like to thank the referee for the positive feedback, according to the provided suggestion we updated figure 1b. We emphasize that despite the presence, of bubbles the relatively large areas (significantly larger than excitation spot) can be easily find in our structure.

Reviewer #2 (Remarks to the Author):

I thank the authors for their rebuttal and revised manuscript. I have the following comments:
1. I agree with Referee 1 about the quality of the data being far from ideal. Nevertheless, the quadrupolar exciton is conclusively observed, so although the quality is unfortunate, the experiment evidence is present for the QX.

We are pleased that the corrections we have made presented results more compelling.

2. I have mixed opinions about the “novelty” or “excitement” of the discovery of QX in a bilayer system. I actually think the discovery is very nice, but I believe it is oversold in the revised manuscript. I guess I differ from Referee 3 who asked for more comments about why this is useful and therefore stronger statements about quantum simulation etc are now more prominent. Perhaps the intro could introduce QX more generally with these applications in mind, rather than making the specific comments about this particular bilayer QX? Softening the claims about the “usefulness” would be more honest in my view. I note also that multiple groups have manufactured the trilayer structures without too much difficulty, so I don't think that the fabrication challenge is such a great selling point.

We agree with referee that strong emphasis on “uniqueness” of bilayer presented in the introduction should be balance with more general description of the importance of quadrupolar states. Therefore according to the referee suggestion, we removed the discussion of the potential advantages of bilayer from the introduction, keeping only information that finding quadrupolar excitons in bilayer systems was unexpected. Nevertheless, we keep a short paragraph about potential advantage of bilayer system in the conclusions. This is motivated by the fact that according to our experience and also many of other leading groups the reproducible fabrication of TMDs stacks represents an important challenge [Nature 602, 41–50 (2022), Nature Nanotechnology 18, 572–579 (2023), Nature

Nanotechnology 15, 750–754 (2020)]. In contrast natural bilayers characterizes with large area homogeneity which is worth to mentioning.

3. The authors have not answered my original query (#2) about the lack of magneto-optics experiments which allows more clear assignment of the spin and valley configurations of the excitons. At the moment, I believe the authors have made a significant error in their interpretation due to the lack of such an experiment: their entire model and simulation rests on the claim that the QX arise from spin singlet states and the IX is a spin triplet. This is directly in contradiction to Nature Comms 15 4377 (2024), which provides strong experimental evidence that the IX is a spin singlet. The authors entire model / simulation is based on false starting point, as far as I can ascertain.

Our results are not contradictory to those reported in *Nature Communications* **15**, 4377 (2024). Rather, they extend the findings by demonstrating that part of the interlayer transition remains in a dipolar state (as also observed in our work), while another part transforms into quadrupolar excitons. While we acknowledge that the effective nature of our model does not allow for a definitive differentiation between a singlet or triplet origin, we provide arguments supporting a singlet origin for the QX transition:

1. Oscillator strength evidence:

From the spectra and false-color maps, the QX features are more pronounced (stronger) than the IX features, which suggests a higher oscillator strength. This behavior aligns with expectations for a singlet transition.

2. Spin-conserving interaction:

The mixing is assumed to arise from Coulomb interactions, which conserve spin. In a bilayer system, the mixing of spin-triplet states would not be spin-conserving and, therefore, should be much weaker. Consequently, singlet states are expected to transform into quadrupolar states.

However, we cannot exclude a contribution from triplet states and hope that our work will trigger future theoretical and experimental studies to further investigate the impact of triplet and singlet states on the emergence of quadrupolar states in TMD bilayers.

Furthermore we want to emphasise that magneto-optical investigations can also yield inconclusive results. For instance, in the study mentioned by the referee, the experimentally determined g-factor for dipolar interlayer transitions is approximately 12. It is important to note that a straightforward analysis of orbital, valley, and spin contributions reveals that the difference between singlet and triplet states is 4 (due to differing spin configurations), with the triplet states exhibiting a higher absolute g-factor [Nano Lett. 2020, 20, 1, 694–700, Phys. Rev. Lett. 123, 027401]. Specifically, the g-factors for singlet and triplet IX transitions in bilayer MoSe₂ are expected to be ~10 and ~14, respectively [*Nature Communications* **15**, 4377 (2024)]. Therefore, the experimental g-factor of ~12 lies between these values, further complicating a definitive assignment.

Taking all of the above into account, in the revised version of the manuscript, we present both scenarios as possible. We provide arguments which make singlet interlayer excitons a more plausible origin of the QX. However, we have softened our statements to avoid excluding the alternative possibility:

“...We initially focus on the exchange coupling to explain the nonlinear shift of the Q1 and Q2 transitions. We assume that J only mixes the 1s interlayer exciton states of the spin-singlet configuration IXs ($\uparrow\uparrow$, $\downarrow\downarrow$) and opposite dipole moments. This tentative assumption is motivated by the more pronounced signature of Q-states compared to IX features, which suggests a higher oscillator strength characteristic for singlet transition. Moreover, the mixing is assumed to stem from the Coulomb interaction, which is a spin-conserving interaction. Such mixing between the necessary spin-triplet states would not be spin-conserving in a bilayer system. Nevertheless, qualitatively similar quadrupole formation could be expected assuming an equally efficient coupling between interlayer triplet states...”

“...According to the presented analysis the IX states ($IX\uparrow\downarrow/IX'\downarrow\uparrow$) with resonance around 90 meV above the A1s at $E_z = 0$ V/nm are transitions originating from optically bright spin-triplet states.⁶² Due to the lack of coupling J these preserve their dipolar character exhibiting a linear Stark shift. Here we note that the opposite assignment of the dipolar transition origin can also be found.⁵⁴ Unfortunately our model does not allow for a definitive differentiation between a singlet or triplet origin of quadrupolar states. Importantly the singlet or triplet nature of the interlayer is not essential to interpret the quadratic shift of Q1 and Q2 transitions....”

We also modified the introduction and the conclusion parts to avoid definitive statements about the singlet or triplet origin of quadrupolar transitions.

Reviewer #3 (Remarks to the Author):

I reviewed the author's response and manuscript revisions. The manuscript has been improved considerably and a transparent description of the experimental method and data analysis has been included. I agree with Reviewer 2 that magneto-optical experiments or other additional evidence could reduce uncertainty in the assignment of spin configurations and the modelling, which would constitute an important improvement. Nevertheless, I believe the main point of the manuscript, the observation of quadrupolar excitons in MoSe2 bilayer, is compelling and I therefore recommend the manuscript for publication in Nature Communications.

We would like to thank referees for their supporting comments.